# Assessment of Oxidative Stress Markers in Hypertensive Patients under the Use of Renin-Angiotensin-Aldosterone Blockers

**DOI:** 10.3390/antiox12040802

**Published:** 2023-03-25

**Authors:** Nestor Vazquez-Agra, Ana-Teresa Marques-Afonso, Anton Cruces-Sande, Estefania Mendez-Alvarez, Ramon Soto-Otero, Jose-Enrique Lopez-Paz, Antonio Pose-Reino, Alvaro Hermida-Ameijeiras

**Affiliations:** 1Department of Internal Medicine, University Hospital of Santiago de Compostela, 15706 A Coruña, Spain; 2Department of Psychiatry, Radiology, Public Health, Nursing and Medicine, University of Santiago de Compostela, 15782 A Coruña, Spain; 3Department of Biochemistry and Molecular Biology, Faculty of Medicine, University of Santiago de Compostela, 15782 A Coruña, Spain

**Keywords:** antioxidants, thiobarbituric acid reactive substances, reduced thiols, renin-angiotensin-aldosterone system, RAAS blockers, chronotherapy, blood pressure, dipper

## Abstract

As in other fields, chronotherapy applied to arterial hypertension (AHT) may have implications on oxidative stress. We compared the levels of some redox markers between hypertensive patients with morning and bedtime use of renin-angiotensin-aldosterone system (RAAS) blockers. This was an observational study that included patients older than 18 years with a diagnosis of essential AHT. Blood pressure (BP) figures were measured using twenty-four-hour ambulatory BP monitoring (24-h ABPM). Lipid peroxidation and protein oxidation were assessed using the thiobarbituric acid reactive substances (TBARS) and reduced thiols assays. We recruited 70 patients with a median age of 54 years, of whom 38 (54%) were women. In hypertensive patients with bedtime use of RAAS blockers, reduced thiol levels showed a positive correlation with nocturnal diastolic BP decrease. TBARS levels were associated with bedtime use of RAAS blockers in dipper and non-dipper hypertensive patients. In non-dipper patients, bedtime use of RAAS blockers was also associated with a decrease in nocturnal diastolic BP. Chronotherapy applied to bedtime use of some BP-lowering drugs in hypertensive patients may be linked to a better redox profile.

## 1. Introduction

Uncontrolled arterial hypertension (AHT) is one of the main causes of morbidity and mortality worldwide. Twenty-four-hour Ambulatory blood pressure monitoring (24-h ABPM) is a better approach than isolated office measurements both for the diagnosis of AHT and prediction of hypertension-mediated organ damage (HMOD) and cardiovascular disease (CVD). Additionally, high nocturnal blood pressure (BP) levels and blunted nocturnal BP decrease are associated with higher cardiovascular risk (CVR) [1].

Inflammation, oxidative stress, turbulent blood flow, and arterial wall stress are central factors in the development of endothelial dysfunction and arterial stiffness in hypertensive patients. The excess of free radicals and reactive oxygen species (ROS) derived from multiple metabolic reactions favors redox imbalance. Oxidative processes can easily spread to major organic molecules with harmful effects at multiple levels [2].

ROS-mediated lipid oxidation enhances the formation of lipid peroxides and lipid peroxidation end products. Lipid peroxidation processes propagate to cell membrane lipids leading to fluidity, permeability, and trans-membrane protein impairment. Furthermore, malondialdehyde (MDA) and 4-hydroxinonenal (4-HNE) are aldehyde end products of lipid peroxidation with pronounced mutagenic and carcinogenic effects [3]. In a pro-oxidant environment, some functional groups in proteins are susceptible to oxidation reactions. Increased levels of carbonyl groups and decreased concentrations of reduced thiols are related to protein structural alterations and enzyme dysfunction [4].

Knowledge of the pathophysiology of BP control has promoted the development of numerous groups of BP-lowering drugs. Although with their respective targets, all of them decrease BP, HMOD, cardiovascular events, and mortality [1]. Several clinical trials have also shown that chronotherapy applied to pre-hypertensive and hypertensive patients may be beneficial. Bedtime use of BP-lowering drugs in some groups of hypertensive individuals has improved the control of 24 h-ABPM indices, reduced the incidence of cardiovascular events, and decreased mortality. Chronotherapy could also be of value in some comorbid populations with a higher prevalence of abnormalities in the circadian BP profile [5].

Blood-pressure-lowering drugs blocking the renin-angiotensin-aldosterone system (RAAS) in monotherapy or combination schemes are the central element in the management of AHT. Additionally, angiotensin II receptor blockers (ARBs), angiotensin-converting enzyme inhibitors (ACEIs), and mineralocorticoid receptor antagonists (MRAs) have shown pleiotropic effects that contribute to the control of BP figures [6,7].

Some studies have also found that BP decreases in hypertensive patients using RAAS blockers are accompanied by lower levels of some redox markers [8]. As in other fields, chronotherapy applied to bedtime use of some BP-lowering drugs may have implications on oxidative stress that have not yet been explored. Therefore, we compared the levels of plasma lipid peroxides and reduced thiols between hypertensive patients with morning-time and bedtime use of RAAS blockers in monotherapy.

## 2. Materials and Methods

### 2.1. Study Design and Participants

This was an observational study conducted in the Cardiovascular Risk Unit belonging to the Department of Internal Medicine at the University Clinical Hospital of Santiago de Compostela (A Coruña, Spain). Patients were recruited between March and June 2022.

We included patients older than 18 with a diagnosis of essential AHT and under the use of RAAS blockers in monotherapy. Smoking abuse, risky alcohol consumption (more than 10 g and 20 g per day in women and men, respectively), and concomitant chronic diseases were exclusion criteria. 

### 2.2. Parameters of 24-h ABPM Collection

Patients underwent 24-h ABPM using an electronic (oscillometric) upper arm cuff device (Space-Labs 90207^®^ device; Space-Labs Inc., Redmond, WA, USA) validated according to the STRIDE blood pressure protocol that is recommended by the European Society of Arterial Hypertension (ESH). Ambulatory BP was recorded every 20 min during the day and every 30 min during the night. The evaluation of day- and night-time periods was performed according to the patient’s report [9]. All patients underwent 24-h ABPM once at the time of the enrollment visit.

The following indices were available from the 24-h ABPM recordings: average 24-h SBP (24-hSBP) and DBP (24-hDBP), average daytime SBP (dSBP) and DBP (dDBP), average nocturnal SBP (nSBP) and DBP (nDBP). Nocturnal SBP decrease and nDBP dipping were calculated as the percentage (ratio between (a) daytime index minus night-time index and (b) the daytime index) and absolute (daytime index minus night-time index) differences of the individual parameters. A non-dipper pattern was defined as a decrease in nSBP and/or nDBP of lower than 10% compared with the average daytime values. A dipper pattern was defined as a decrease in nSBP and nDBP equal to or higher than 10% compared with the average daytime values [9].

### 2.3. Clinical and Laboratory Baseline Variables

We collected information on age, sex, and alcohol consumption. Body mass index (BMI) was calculated with the weight (kg) and height (m) data and expressed in kg/m^2^. WC was measured over both iliac crests with a qualified tape. The degree of therapeutic compliance was assessed using the Morisky-Green questionnaire [10]. Blood samples were obtained at 08:00 a.m. following overnight fasting of 12 h and ensuring a minimum of 12 h since the last use of RAAS blockers. We evaluated serum levels of fasting plasma glucose (FPG), creatinine, uric acid, total cholesterol (TC), triglycerides (TG), and total proteins. The glomerular filtration rate (eGFR) was estimated using the MDRD equation [11].

### 2.4. Assessment of Thiobarbituric Acid Reactive Substances

We assessed the levels of plasma lipid peroxides by the quantification of thiobarbituric acid reactive substances (TBARS) levels. Blood samples were collected in tubes with ethylenediaminetetraacetic acid (EDTA) and centrifuged less than 1 h after extraction at 1000 G and 4 °C for 10 min. Thiobarbituric acid (TBA) forms an adduct with some aldehydes under acidic conditions and at high temperatures. The TBA-adduct complex releases radiation in the visible light spectrum so that its absorbance is easily quantifiable by colorimetric methods. We made a mixture of 1% TBA (reference code: T5500-25G, ≥98%, Sigma-Aldrich^®^, St. Louis, MO, USA) in 20% acetic acid solution, adjusted to pH = 3 with sodium hydroxide following the protocol of Ohkawa et al. We added a 10-millimolar (mM) Butylhydroxytoluene (BHT) solution in pure ethanol to protect the sample from auto-oxidation processes and in vitro lipid peroxidation [12]. 

We performed a standard calibration with a colorimetric curve using MDA (reference code: 8207560250, ≥99%, Sigma-Aldrich^®^, St. Louis, MO, USA) and achieved a coefficient of determination higher than 98%. The mixture was heated at 90° Celsius for 30 min, cooled to −20° Celsius for 10 min, and finally centrifuged at 4000 G for another 10 min. Finally, 250 microliters (µL) of the test and calibration samples were analyzed in triplicate using an Asys UVM-340 analyzer (Biochrom^®^, Cambridge, UK) at a wavelength of 530–540 nm. Given the influence of serum lipoprotein (Lp) levels on the assessment of TBARS levels, we adjusted the concentrations to Lp levels, and the results were expressed as nanomoles (nmol) per milligram (mg) (nmol/mg) of Lp [13].

### 2.5. Assessment of Reduced Thiols

We assessed the levels of plasma protein oxidation by the quantification of reduced thiols. The Ellman technique uses 5,5-dithio-bis-(2-nitrobenzoic) acid (reference code: D8130, ≥99%, Sigma-Aldrich^®^, St. Louis, MO, USA) as a reagent to form a compound with the sulfhydryl groups of some amino acid residues, producing a colored pigment that can be measured colorimetrically and whose absorbance is directly proportional to the levels of reduced thiols and inversely proportional to the degree of protein oxidation. We prepared a buffer of 0.1 molar (M) sodium triphosphate and 1 mM EDTA adjusted to pH = 8 with hydrochloric acid and performed a standard calibration with a colorimetric curve using L-cysteine hydrochloride (reference code: C7880, ≥98%, Sigma-Aldrich^®^, St. Louis, MO, USA) 1.5 mM diluted in the buffer. The coefficient of determination was higher than 98%. Finally, 250 µL of buffer, 5 µL of Ellman’s reagent, and 25 µL of calibrated or test sample were analyzed in triplicate using an Asys UVM-340 analyzer (Biochrom^®^, Cambridge, UK) at a wavelength of 412 nm and after 20 min of solution resting. The levels of reduced thiols were expressed in mM [14,15].

### 2.6. Ethics Statement

This study was conducted following the ethical principles of the Declaration of Helsinki and standards of good practice (NBP) in research of the Galician (Spain) health service (SERGAS). All patients who consented to participate provided written informed consent. The Research Ethics Committee of Santiago-Lugo approved the study protocol (code number 2021/401).

### 2.7. Statistical Analysis

Statistical analysis was conducted using SPSS 22.0 statistical software (SPSS Inc., Chicago, IL, USA). We conducted a descriptive analysis in which qualitative and quantitative variables were expressed as a number (percentage) and median (interquartile range), respectively. In the univariate analysis, qualitative and quantitative variables were compared using the chi-square and Mann–Whitney U test, respectively. We performed a linear correlation analysis between some indices of nocturnal BP decrease and the oxidative stress markers. The results were expressed as the Pearson or Spearman correlation coefficients as appropriate.

An analysis of interaction and confounding phenomena was performed for the variables that reached statistical significance in the univariate analysis before constructing the multivariate models. Finally, we developed binary logistic regression models to evaluate the association between TBARS levels and bedtime use of RAAS blockers in dipper and non-dipper hypertensive patients. The interaction terms that reached statistical significance and the factors that modified the effect measure for the variables of interest by more than 10% were included in the final multivariate models. The sample size was calculated using the Epidata© software (http://www.epidata.dk/) and considering a 95% confidence level with a statistical power of not less than 80% and a standardized mean difference in TBARS and reduced thiol levels between the groups to be detected of no less than 1. We considered a *p*-value of less than 0.05 as the threshold for statistical significance [16].

## 3. Results

We recruited 70 patients with a median age of 54 years, of whom 38 (54%) were women. A total of 29 (41%) patients had a non-dipper BP profile, and more than 90% of the patients showed good therapeutic adherence. All the clinical results are shown in Table 1. The 24-h, daytime, and night-time BP indices showed median values in the range of adequate BP control. All BP figures are summarized in Table 2, and all BP figures at every 1 h interval are also provided in Appendix A. The results concerning laboratory variables, including TBARS and reduced thiol levels, are summarized in Table 3.

### 3.1. Comparisons between Hypertensive Patients with Morning and Bedtime Use of RAAS Blockers

Hypertensive patients with bedtime use of RAAS blockers were older and had a higher BMI than individuals with morning-time use, although there were no differences in sex. The prevalence of dipper and non-dipper profiles in the groups was similar, and the frequency of alcohol consumption, therapeutic compliance, and BP-lowering drugs was also similar. All the results are provided in Table 1. We found no differences in BP indices between the groups, as shown in Table 2 and Figure 1. 

Overall, nocturnal DBP decrease was positively correlated with the levels of reduced thiols (Rho = 0.242 *p* = 0.043) in hypertensive patients. However, the correlations between TBARS levels and nocturnal BP decrease did not reach statistical significance. In patients with bedtime use of RAAS blockers, we also found a correlation between the levels of reduced thiols and the percentage of nDBP decrease that we did not find in individuals with morning-time dosage. The results are provided in Figure 2.

Regarding laboratory variables, we found higher TG levels in the group of patients with bedtime use. TBARS levels were lower in hypertensive patients with bedtime use of RAAS blockers, and we did not find differences in reduced thiol levels between the groups. All the results are provided in Table 3 and Figure 1.

### 3.2. Comparisons between Dipper Patients with Morning and Bedtime Use of RAAS Blockers

Dipper patients with bedtime use of RAAS blockers were older than individuals with morning-time use, and we did not find differences in sex and anthropometric indices between the groups. The results did not show differences in alcohol consumption, therapeutic compliance, and BP-lowering drugs between the groups. All the results are provided in Table 1. We found no differences in BP indices between the groups, as shown in Table 2 and Figure 1. 

Regarding laboratory variables, we found higher TG levels in the group of patients with bedtime use of RAAS blockers. TBARS levels were lower in dipper patients with bedtime use of RAAS blockers, and we did not find differences in reduced thiol levels between the groups. All the results are provided in Table 3 and Figure 1.

### 3.3. Comparisons between Non-Dipper Patients with Morning and Bedtime Use of RAAS Blockers

Non-dipper patients with morning- and bedtime use of RAAS blockers were comparable in age, sex, and anthropometric indices. The prevalence of alcohol consumption, therapeutic adherence, and BP-lowering drugs was similar between the groups. All the results are provided in Table 1. Twenty-four-hour DBP, dDBP, and nDBP were lower in non-dipper patients under bedtime use of RAAS blockers, as shown in Table 2 and Figure 1. 

As for laboratory variables, we found higher TG levels in the group of patients with bedtime use. TBARS levels were lower in non-dipper patients under bedtime use of RAAS blockers. The levels of reduced thiols did not show relevant results (Table 3 and Figure 1). 

Although TBARS levels and DBP indices were lower in non-dipper patients who were under bedtime use of RAAS blockers, the correlation between TBARS concentrations and 24-h DBP (Rho = −0.085, *p* = 0.661), dDBP (Rho = −0.086, *p* = 0.656), and nDBP (Rho = −0.124, *p* = 0.522) did not reach statistical significance. 

### 3.4. Multivariate Analysis: Association between TBARS Levels and Bedtime Use of RAAS Blockers

Table 4 shows the binary logistic regression models for the association between TBARS levels and bedtime use of RAAS blockers. After adjustment for age, sex, BMI, WC, nocturnal BP indices, and eGFR, bedtime use of RAAS blockers was associated with lower levels of TBARS in hypertensive patients with dipper and non-dipper BP profiles. In the dipper group, age was also a variable in the model, whereas, in non-dipper patients, bedtime use of RAAS blockers was also associated with lower nDBP levels.

## 4. Discussion

The main results are summarized as follows: (I) Hypertensive patients under bedtime use of RAAS blockers had lower TBARS levels than patients with morning-time use; (II) Reduced thiol levels showed a positive correlation with nocturnal DBP decrease in individuals with bedtime use of RAAS blockers; (III) In patients with a non-dipper BP profile and bedtime use of RAAS blockers, lower TBARS levels were concomitant with a decrease in the diastolic BP indices; (IV) After multivariate analysis, TBARS levels were associated with bedtime use of RAAS blockers both in dipper and non-dipper hypertensive patients. In non-dipper patients, bedtime use of RAAS blockers was also associated with a decrease in nocturnal diastolic BP.

The literature supports that patients with controlled AHT have a better redox profile than those individuals with out-of-normal BP figures. Additionally, multiple studies have suggested that improved redox status contributes to maintaining adequate BP control. Oxidative stress and endothelial dysfunction are essential conditions for developing vascular tone abnormalities that influence BP levels and the circadian rhythm of BP [17,18,19].

The relationship between oxidative stress, RAAS, and AHT has been addressed in multiple studies. Angiotensin II-mediated activation of the angiotensin II type 1 receptor (AT1) increases free radicals production in target cells and activates some inflammatory pathways such as NF-KB. In turn, the presence of a pro-oxidant milieu has been associated with up-regulation in the number of receptors and cellular response mediated by AT1 [20].

The role of bedtime use of RAAS blockers on the levels of lipid peroxides and its end products in hypertensive patients is still a poorly explored concern. Plasma lipid peroxides increase with age and certain comorbid conditions, such as diabetes mellitus, hyperlipidemia, AHT, and chronic kidney disease, among others. However, although hypertensive patients with bedtime use of RAAS blockers were older and had a higher BMI and more abnormalities in their lipid profile, TBARS levels were markedly lower in this group of patients. Further studies would be necessary to assess whether this could be a general finding for patients with AHT, a specific feature for some groups of hypertensive patients, or a phenomenon depending on the therapeutic group, among other factors [21,22,23].

The results showed a decrease in TBARS levels in both dipper and non-dipper hypertensive patients. One possible interpretation may be that the decrease in lipid peroxide levels is independent of the circadian BP profile and a common denominator in well-controlled hypertensive patients under bedtime use of RAAS blockers in monotherapy. However, the reality could be much more complex, and further evidence is needed.

We also found an association between nDBP figures and TBARS levels with bedtime use of RAAS blockers in non-dipper patients, but not in the dipper group. This phenomenon could have several readings since chronotherapy with bedtime use of some therapeutic regimens deactivates mechanisms favoring high nocturnal BP levels (such as the RAAS), improves sleep dynamics, and controls the morning surge of BP in hypertensive patients with blunted nocturnal BP decrease. Furthermore, it is a fact that non-dipper patients on bedtime use of BP-lowering drugs also have a greater chance of improvement in nocturnal BP figures than dipper individuals [24,25,26,27,28].

As for reduced thiols, we did not find differences between the comparison groups. However, hypertensive patients with bedtime use of RAAS blockers showed a joint variation of nDBP decrease with reduced thiol levels that we did not find in patients with morning-time use. Some studies in AHT, circadian BP profile, and redox markers support a relationship between the levels of protein oxidation markers and BP figures, while reduced thiol concentrations have also been correlated with the nDBP decrease [19,29]. However, the influence of chronotherapy on the interrelationship between protein oxidation markers and BP indices remains unknown.

According to the evidence on circadian rhythms and oxidative stress, the levels of reduced thiols are higher during the night-time period, while potentially pro-oxidant and BP-raising factors, such as some components of the RAAS system, exhibit increased activity around the middle of night-time sleep. Therefore, the correlation between reduced thiols and nocturnal BP decrease in patients with bedtime use of RAAS blockers may be a possible scenario, although further studies are needed [20,30,31].

### Limitations and Strengths

This was an observational and single-center study conducted in real clinical practice. The patients were middle-aged, white European individuals without concomitant chronic diseases, so the results may not be applied to other populations. The small sample size could also be a limitation since, in other samples from the target population, some relevant clinical variables such as age, sex, and anthropometric indices could have had a different impact on the results. Although we performed a systematic collection of variables, some factors related to both blood pressure and redox status may not have been included. The classical distinction between dipper and non-dipper patients based on the 10% nocturnal BP decrease is a more complex reality nowadays with the presence of some other circadian BP patterns with their peculiarities, such as riser or very dipper BP profiles [32]. Two single markers of oxidative stress may provide biased information on redox status, and the measurement of more oxidative processes on other organic molecules, such as nucleic acids, would be more accurate in the evaluation of oxidative stress. 

The assessment of plasma TBARS by colorimetric methods, although a very sensitive approach, has lower specificity than other techniques, such as chromatography. The time between the last dose intake and blood sampling was undeniably different in patients with morning and bedtime use of RAAS blockers so certain pharmacokinetic conditions could influence the levels of redox markers [33]. The best option to confirm or refute the main hypothesis of the study would be an experimental design to contrast the differences in redox markers between morning- and bedtime use of BP-lowering drugs in all patients. Therefore, the study should be seen as a first approximation, and further evidence is needed.

## 5. Conclusions

Chronotherapy applied to the bedtime use of renin-angiotensin-aldosterone system blockers in hypertensive patients may help decrease lipid peroxidation independently of the patient’s circadian blood pressure pattern. The Bedtime use of some blood-pressure-lowering drugs may be a way to improve the beneficial synergy between some antioxidant mechanisms, such as reduced thiols, and blood pressure indices, through the attenuation of closely related mechanisms favoring oxidative stress and blood pressure raising that are exacerbated during the night-time. The results of this study could be the starting point of a line of research on the role of chronotherapy in arterial hypertension, abnormalities in circadian BP profile, and oxidative stress.

## Figures and Tables

**Figure 1 antioxidants-12-00802-f001:**
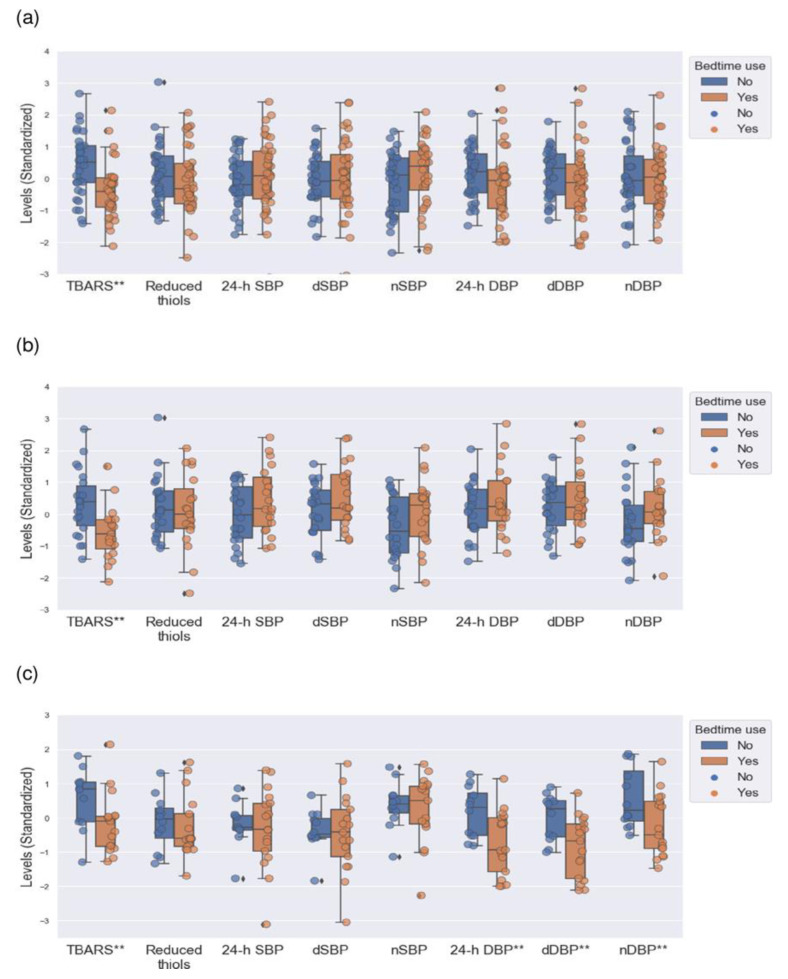
Comparison of redox markers levels and 24-h ABPM indices between patients with morning and bedtime use of RAAS blockers. (**a**) Hypertensive patients. (**b**) Dipper hypertensive patients. (**c**) Non-dipper hypertensive patients. Box-and-whisker diagram. The horizontal line of the box represents the median, and the upper and lower box edges represent the 3rd and 1st quartiles (Q), respectively. The edges of the upper and lower whiskers represent the Q3 + 1.5 × interquartile range and Q1 − 1.5 × interquartile range, respectively. Data represented by diamonds are outliers. The point cloud represents the distribution of the sample values. The non-standardized results expressed as median and interquartile range are shown in Table 2 and Table 3. TBARS levels, reduced thiol concentrations, and blood pressure indices were measured in nmol/mg Lp, mmol/L, and mmHg, respectively. Terms: 24-h ABPM–Twenty-four-hour ambulatory blood pressure monitoring; RAAS–Renin-angiotensin-aldosterone system; SBP–Systolic blood pressure; 24-hSBP–Average SBP over 24 h; dSBP–Average SBP during the day; nSBP–Average SBP during the night; DBP–Diastolic blood pressure; 24-hDBP–Average diastolic BP over 24 h; dDBP–Average DBP during the day; nDBP–Average DBP during the night; nmol/mg Lp–Nanomol per milligram of serum lipoprotein; mmol/L–Millimolar; mmHg–Millimeter of mercury. ** refers to a *p*-value of lower than 0.05.

**Figure 2 antioxidants-12-00802-f002:**
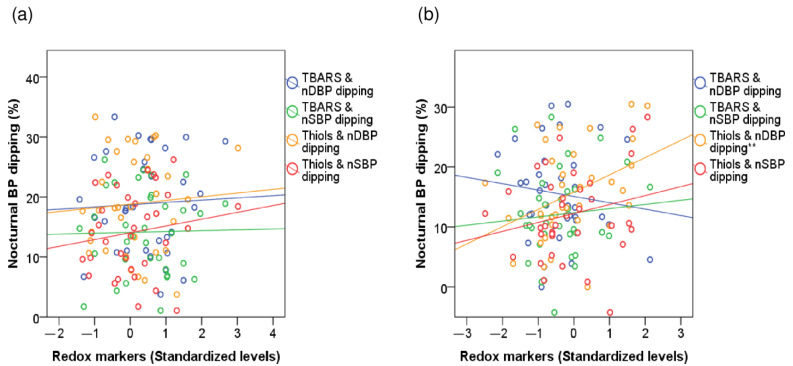
Correlations between redox markers and nocturnal BP decrease. (**a**) Patients with morning-time use of RAAS blockers. TBARS and nDBP dipping: Rho = −0.008, *p* = 0.965. TBARS and nSBP dipping: Rho = −0.032, *p* = 0.859. Reduced thiols and nDBP dipping: Rho = 0.052, *p* = 0.768. Reduced thiols and nSBP dipping: Rho = 0.157, *p* = 0.375. (**b**) Patients with bedtime use of RAAS blockers. TBARS and nDBP dipping: Rho = −0.157, *p* = 0.360. TBARS and nSBP dipping: Rho = −0.004, *p* = 0.983. Reduced thiols and nDBP dipping: Rho = 0.416, *p* = 0.012. Reduced thiols and nSBP dipping: Rho = 0.210, *p* = 0.220. BP−Blood pressure. RAAS−Renin-angiotensin-aldosterone system. TBARS−Thiobarbituric acid reactive substances. nDBP−Nocturnal diastolic blood pressure. nSBP−Nocturnal systolic blood pressure. Rho−Spearman’s correlation coefficient. ** refers to a *p*-value of less than 0.05.

**Table 1 antioxidants-12-00802-t001:** Clinical variables. Comparisons between morning and bedtime use of RAAS blockers in dipper and non-dipper hypertensive patients.

Variables	Total n = 70	All Patients ^1^ n = 70		Dippers ^2^ n = 41		Non-Dippers ^3^ n = 29	
		Morning-Time n = 34	Bedtime n = 36	Morning-Time n = 22	Bedtime n = 19	Morning-Time n = 12	Bedtime n = 17
Age (years) †	54 (14)	49 (17)	57 (13) ^a^	48 (18)	58 (12) ^c^	55 (21)	57 (15)
Sex (women) ‡	38 (54)	20 (59)	18 (50)	13 (59)	7 (37)	7 (58)	11 (65)
Weight (kg) †	75 (27)	70 (34)	77 (25)	70 (33)	75 (22)	70 (29)	80 (27)
Height (cm) †	164 (12)	164 (10)	164 (16)	161 (12)	170 (16)	165 (10)	163 (11)
BMI (kg/m^2^) †	29 (6)	27 (7)	29 (6) ^b^	27 (8)	29 (5)	28 (5)	31 (7)
WC (cm) †	101 (20)	98 (29)	104 (13)	93 (29)	103 (26)	101 (31)	105 (18)
Alcohol intake (yes) ‡	11 (16)	5 (15)	6 (17)	1 (5)	2 (10)	4 (33)	4 (24)
Non-dipper profile (yes) ‡	29 (41)	12 (35)	17 (47)				
Compliant patients (yes) ‡	66 (94)	32 (94)	34 (94)	21 (96)	17 (90)	11 (92)	17 (100)
ACEIs ‡	17 (24)	8 (24)	9 (25)	3 (14)	4 (21)	5 (41)	5 (30)
ARBs ‡	49 (70)	25 (74)	24 (67)	19 (86)	13 (69)	6 (50)	11 (64)
MRAs ‡	4 (6)	1 (3)	3 (8)	0 (0)	2 (10)	1 (8)	1 (6)

^1^ Comparisons between hypertensive patients with morning and bedtime use of RAAS blockers (^a^
*p =* 0.005 ^b^
*p =* 0.015). ^2^ Comparisons between dipper patients with morning and bedtime use of RAAS blockers (^c^
*p =* 0.008). ^3^ Comparisons between non-dipper patients with morning and bedtime use of RAAS blockers. Comparisons in which the *p*-value was not provided have not reached statistical significance. RAAS–Renin-angiotensin-aldosterone system. BMI–Body mass index. WC–Waist circumference. ACEI–Angiotensin-converting enzyme inhibitor. ARB–Angiotensin II receptor blockers. MRA–Mineralocorticoid receptor antagonistResults expressed as † refer to the median and interquartile range. Results expressed ‡ refer to number and percentage.

**Table 2 antioxidants-12-00802-t002:** Indices of twenty-four-hour ambulatory blood pressure monitoring. Comparisons between morning and bedtime use of RAAS blockers in dipper and non-dipper hypertensive patients.

Variables	Total n = 70	All Patients ^1^ n = 70		Dippers ^2^ n = 41		Non-Dippers ^3^ n = 29	
		Morning-Time n = 34	Bedtime n = 36	Morning-Time n = 22	Bedtime n = 19	Morning-Time n = 12	Bedtime n = 17
24-hSBP (mmHg) †	125 (15)	123 (14)	126 (18)	125 (20)	127 (21)	122 (5)	121 (18)
dSBP (mmHg) †	129 (15)	129 (15)	130 (20)	135 (18)	133 (20)	124 (8)	125 (21)
nSBP (mmHg) †	116 (19)	114 (21)	118 (17)	107 (22)	116 (17)	118 (7)	119 (14)
24-hDBP (mmHg) †	76 (14)	79 (12)	76 (11)	78 (12)	79 (12)	79 (13)	68 (16) ^a^
dDBP (mmHg) †	79 (15)	84 (13)	80 (15)	85 (14)	83 (15)	84 (12)	74 (17) ^b^
nDBP (mmHg) †	66 (13)	66 (13)	67 (13)	63 (11)	67 (9)	69 (16)	62 (13) ^c^

^1^ Comparisons between hypertensive patients with morning and bedtime use of RAAS blockers. ^2^ Comparisons between dipper patients with morning and bedtime use of RAAS blockers. ^3^ Comparisons between non-dipper patients with morning and bedtime use of RAAS blockers (^a^
*p =* 0.014, ^b^
*p =* 0.016, ^c^
*p* = 0.027). Comparisons in which the *p*-value was not provided have not reached statistical significance. RAAS–Renin-angiotensin-aldosterone system. SBP–Systolic blood pressure. Terms: 24-hSBP–Average SBP over 24 h; dSBP–Average SBP during the day; nSBP–Average SBP during the night; DBP–Diastolic blood pressure; 24-hDBP–Average diastolic BP over 24 h; dDBP–Average DBP during the day; nDBP–Average DBP during the night; mmHg–Millimeter of mercury. Results expressed as † refer to the median and interquartile range.

**Table 3 antioxidants-12-00802-t003:** Laboratory variables. Comparisons between morning and bedtime use of RAAS blockers in dipper and non-dipper hypertensive patients.

Variables	Total n = 70	All Patients ^1^ n = 70		Dippers ^2^ n = 41		Non-Dippers ^3^ n = 29	
		Morning-Time n = 34	Bedtime n = 36	Morning-Time n = 22	Bedtime n = 19	Morning-Time n = 12	Bedtime n = 17
FPG (mg/dL) †	96 (21)	95 (17)	96 (26)	95 (20)	97 (26)	97 (26)	95 (23)
Creatinine (mg/dL) †	0.83 (0.2)	0.81 (0.2)	0.89 (0.2)	0.82 (0.3)	0.88 (0.3)	0.81 (0.1)	0.86 (0.2)
eGFR (ml/min) †	81 (20)	85 (17)	78 (23)	86 (18)	79 (19)	84 (15)	74 (25)
Uric acid (mg/dL) †	4.9 (2.4)	4.8 (2.6)	5.0 (2.1)	3.9 (2.1)	5.3 (2.5)	5.3 (2.2)	4.8 (2.0)
Total proteins (g/dL) †	7.3 (0.5)	7.2 (0.6)	7.3 (0.6)	7.2 (0.5)	7.3 (0.5)	7.2 (0.8)	7.2 (0.7)
TG (mg/dL) †	88 (43)	72 (33)	101 (52) ^a^	74 (41)	101 (68) ^c^	72 (19)	96 (51) ^e^
TC (mg/dL) †	183 (47)	180 (45)	191 (49)	180 (49)	200 (64)	178 (44)	182 (35)
TBARS (nmol/mg Lp) †	3.1 (1.6)	3.8 (1.3)	2.8 (1.0) ^b^	3.7 (1.6)	2.6 (1.1) ^d^	4.1 (1.2)	3.1 (1.0) ^f^
Reduced thiols (mmol/L) †	0.61 (0.1)	0.63 (0.1)	0.59 (0.1)	0.63 (0.1)	0.62 (0.1)	0.62 (0.1)	0.57 (0.1)

^1^ Comparisons between hypertensive patients with morning and bedtime use of RAAS blockers (^a^
*p* < 0.001, ^b^
*p* = 0.001). ^2^ Comparisons between dipper patients with morning and bedtime use of RAAS blockers (^c^
*p* = 0.002, ^d^
*p =* 0.004). ^3^ Comparisons between non-dipper patients with morning and bedtime use of RAAS blockers (^e^
*p =* 0.003, ^f^
*p =* 0.049). Comparisons in which the *p*-value was not provided have not reached statistical significance. RAAS–Renin-angiotensin-aldosterone system. FPG–Fasting plasma glucose. eGFR–Estimated glomerular filtration rate. TG–Triglycerides. TC–Total cholesterol. TBARS–Thiobarbituric acid reactive substances. mg–milligram. dL–deciliter. g–gram. nmol/mg Lp–nanomol per milligram of lipoprotein. mmol–millimol. L–liter Results expressed as † refer to the median and interquartile range.

**Table 4 antioxidants-12-00802-t004:** Binary logistic regression models for the bedtime use of RAAS blockers.

Variables	B	*p*-Value	Exp(B)	95%CI	
				Inferior	Superior
Hypertensive patients ^a^					
Age (years)	0.083	0.004	1.087	1.027	1.151
TBARS (nmol/mg Lp)	−1.077	0.001	0.341	0.180	0.645
Patients with a dipper BP profile ^b^					
Age (years)	0.100	0.021	1.106	1.016	1.204
TBARS (nmol/mg Lp)	−1.241	0.009	0.289	0.114	0.735
Patients with a non-dipper BP profile ^c^					
nDBP (mmHg)	−0.159	0.023	0.853	0.743	0.979
TBARS (nmol/mg Lp)	−1.207	0.022	0.299	0.106	0.840

^a^ Number of patients included: 70. Considered variables: age, sex, BMI, WC, nocturnal BP indices, eGFR, and TBARS levels. Model summary: *p*-value (F-test) < 0.001, −2LL = 70.11, R^2^ (Nagelkerke) = 0.425, Overall accuracy = 0.74. ^b^ Number of patients included: 41. Considered variables: age, sex, BMI, WC, nocturnal BP indices, eGFR, and TBARS levels. Model summary: *p*-value (F-test) < 0.001, −2LL = 40.14, R^2^ (Nagelkerke) = 0.442, Overall accuracy = 0.83. ^c^ Number of patients included: 29. Considered variables: age, sex, BMI, WC, nocturnal BP indices, eGFR, and TBARS levels. Model summary: *p*-value (F-test) < 0.002, −2LL = 27.27, R^2^ (Nagelkerke) = 0.458, Overall accuracy = 0.76. RAAS–Renin-angiotensin-aldosterone system. TBARS–Thiobarbituric acid reactive substances. BP–Blood pressure. nDBP–nocturnal diastolic blood pressure. BMI–Body mass index. WC–Waist circumference. eGFR–Estimated glomerular filtration rate.–nmol/mg Lp–nanomol per milligram of lipoprotein. mmHg–millimeter of mercury.

## Data Availability

Data presented in this study are available on request from the corresponding author. In accordance with Article 18.4 of the Spanish Constitution and the Organic Law on Data Protection and Guarantee of Digital Rights (LOPDGDD) of 6 December 2018, the privacy and integrity of the individual will be protected at all times, so anonymous data are available upon reasonable request.

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
