# Peer review of "Assessment of Oxidative Stress Markers in Hypertensive Patients under the Use of Renin-Angiotensin-Aldosterone Blockers"

_antioxidants, 2023, doi:10.3390/antiox12040802_

Round 1
Reviewer 1 Report
General comments:
This manuscript describes experiments characterizing plasm lipid peoxides in dipper and non-dipper hypertensive patients with morning and bedtime use of renin-angiotensin-aldosterone systems blockers in monotherapy. The principal result was that bedtime use of the blockers with blunted nocturnal blood pressure may have positive effects in both the reduction in flood pressure and lipid peroxidation.
Overall, the paper is interesting and timely. This reviewer has only minor comments.
Specific comments:
Abstract and Introduction: Please state a hypothesis.
Figures should be plotted as scatter plots.
Reviewer 2 Report
The manuscript reports a potentially important and interesting data. However, the conclusion is based only on the assessment of plasma TBARS levels by colorimetric method. This method has low specificity (as authors mentioned in: Limitation and strengths) . It seems to me that authors should confirm the presented result by measuring other markers of oxidative stress like: 8-isoprostane, 8-hydroxyguanine and carbonyl groups.
Therefore, I cannot recommended publishing this article in the present form.
Reviewer 3 Report
Response to authors:
The authors demonstrated that bedtime use of renin-angiotensin aldosterone system (RAAS) blockers may have positive effects in both the reduction in blood pressure (BP) and lipid peroxidation [thiobarbituric acid reactive substances (TBARS)]. This manuscript is interesting. However, there are some problems in this manuscript.
(1) In general, circadian BP rhythms are classified as either extreme dipper, dipper,
nondipper, or riser patterns when the night-to-day ratio of systolic BP is <0.80, 0.80 to <0.90, 0.90–1.00, and >1.00, respectively. However, the authors defined the circadian rhythms as dipper pattern and non-dipper pattern. The authors should indicate the accuracy more clearly.
(2) The authors indicated the clinical and laboratory baseline variables. However, I
think that they are not sufficient. Essential data such as height, body weight, body mass index as well as estimated glomerular filtration rate (eGFR) are necessary to show the patients’ background. Therefore, the authors should indicate them.
(3) The authors used the variables that were significantly different between morning
and bedtime use of RAAS blockers in multivariate analysis. However, in general, it is necessary to use variables such as age, sex and physique in multivariate analysis. Therefore, the authors should add the variables in multivariate analysis.
(4) It is not clear whether lower TBARS levels depend on the effect of lowering BP or
RAAS inhibitory effect. The authors should indicate them as clear as possible.
(5) It is not clear whether the markers of lipid peroxidation other than TBARS show the same results. The authors should show the results of the markers of lipid peroxidation other than TBARS.
Reviewer 4 Report
Halmid-Ameijeiras et al. reported the lipid peroxidation in hypertensive patients with RAAS inhibitor use. The results are interesting. However, precise protocol is missing. Just the comparison of blood pressure in patients under bedtime and morning time use is not enough. To compare the differences between morning time and bedtime use, 24-hr blood pressure measurements with morning time use and bedtime use should be performed in all patients.
Major comments
The protocol of the study is not clear. When this study was performed? 24-hr blood pressure measurements were performed just once in each patient?
Figure 1 (probably because I could not find Figure 1 in page 5) is standardized. It is difficult to understand the difference in level. Figures of usual 24-hr measurements of blood pressure at every 1 hr should be presented. It is easy to understand dipper and non-dipper. It is not enough to determine the differences in blood pressure between bedtime use and morning time use in different patients. The differences between morning time and bedtime use should be examined by 24 he ABPM in all patients. If redox imbalance is different with bedtime and morning time use of RAAS blockers not by different patients but by all patients, the data would be quite useful.
Minor comments
Abstracts is too long.
What kinds of RAAS inhibitors were used. There are differences among ARB and ACE-I. Gathering the data form patients with ARB and ACE-I would be reasonable. But including the data from aldosterone receptor blocker is not acceptable. If 24 hr ABPM is measured both in morning time use and bedtime use in all patients, precise description of RAAS blockers is not necessary.
Round 2
Reviewer 2 Report
The authors significantly modified the manuscript. However, the scientific and practical value of the revised manuscript is still very low. I agree with authors concusion " The results of this study could be the starting point" (no more) for study " on the role of chronotherapy in arterial hypertension, abormalities in circardian BP profile, and oxidative stress.
Overall, the presnted results in revised version do not allow me to recommed the manuscript for publication in Antioxidants.
Reviewer 3 Report
The authors responded to my concerns adequately. Therefore, there are no special problems in this revised manuscript.
Reviewer 4 Report
The revised version by Vazquez-Agra, et al. improved very much. I have no further comments.
